# Performance Comparison of Procalcitonin, Delta Neutrophil Index, C-Reactive Protein, and Serum Amyloid A Levels in Patients with Hematologic Diseases

**DOI:** 10.3390/diagnostics13071213

**Published:** 2023-03-23

**Authors:** Jooyoung Cho, Jong-Han Lee, Dong Hyun Lee, Juwon Kim, Young Uh

**Affiliations:** 1Department of Laboratory Medicine, Yonsei University Wonju College of Medicine, Wonju 26426, Republic of Korea; 2Center for Precision Medicine and Genomics, Wonju Severance Christian Hospital, Wonju 26426, Republic of Korea

**Keywords:** procalcitonin, delta neutrophil index, C-reactive protein, serum amyloid A, hematologic disease, infection, mortality

## Abstract

(1) Background: We compared the diagnostic and prognostic performance of serum amyloid A (SAA), procalcitonin (PCT), delta neutrophil index (DNI), and C-reactive protein (CRP) in patients with hematologic diseases; (2) Methods: We retrospectively collected the remaining serum samples from patients with hematologic diseases, analyzed their clinical data, and measured the levels of PCT, DNI, CRP, and SAA. Performances for infection diagnosis were evaluated using a receiver operating characteristic curve analysis, and 90-day mortality was analyzed using Kaplan–Meier estimation; (3) Results: The levels of all markers were significantly higher in the infected group (*N* = 27) than those in the uninfected group (*N* = 100) (*p* < 0.0001 for all markers). The areas under the curve for diagnosing infection for PCT, DNI, CRP, and SAA were 0.770, 0.817, 0.870, and 0.904, respectively. Increased PCT levels were associated with higher mortality (*p* = 0.0250); this association was not observed with other examined markers; (4) Conclusions: CRP and SAA exhibited greater discriminative power for infection than PCT. However, only PCT levels were positively associated with 90-day mortality. Herein, we evaluated the diagnostic performance of the four markers. Additional studies are needed to confirm the findings of the present study and validate the potential of these markers in clinical practice.

## 1. Introduction

Inflammatory reactions occur as a response to stimuli or disease progression and have been associated with disease activity or prognosis in various diseases, including trauma, injury, surgery, malignancy, and rheumatoid arthritis, as well as cardiac, renal, and hepatic diseases [1,2]. Inflammation has been associated with the etiology of several hematologic diseases, including hematologic malignancies [3,4]; however, it remains challenging to determine whether the presence of fever in patients with hematologic diseases can be attributed to an infection, malignancy, or the disease itself. Therefore, early and differential diagnoses of inflammatory diseases are essential for initiating proper treatment and suitable patient management [2,3].

For decades, laboratory biomarkers have been employed to diagnose inflammatory diseases. Procalcitonin (PCT), a precursor of the hormone calcitonin, is rapidly produced by thyroid C cells in response to proinflammatory stimulation [5,6]. PCT levels are known to increase significantly in response to bacterial endotoxins, contributing to the early detection and diagnosis of inflammation of bacterial origin; however, elevated PCT levels are mainly limited to bacterial infection [7]. The delta neutrophil index (DNI), introduced in previous studies, is a biomarker that reflects circulating immature granulocytes. Immature, left-shifted granulocytes have been documented as indicators of sepsis or septic shock [8,9,10].

Acute-phase reactants (APRs) are proteins with elevated concentrations during the early phase of inflammatory conditions with high sensitivity, playing a major role in the early detection of inflammatory diseases [2,11]. However, distinguishing the underlying cause of inflammation, including infection, based on the increasing APR pattern can pose a challenge [6]. C-reactive protein (CRP) is one of the most popular APRs, routinely used in several clinical laboratories [6,12]. Like CRP, serum amyloid A (SAA) is an APR produced by hepatocytes in the liver [13]. The reference range for SAA is typically below 10 μg/mL; however, it is generally <1 μg/mL in healthy individuals [2,14]. Blood concentrations of SAA and CRP are markedly low in healthy individuals but can increase up to 1000-fold within 24 h after the onset of inflammatory stimulation, decreasing in a similar pattern after the resolution of the underlying inflammatory condition [11,15]. Although CRP and SAA exhibit parallel increase and decrease patterns, some previous studies have reported SAA to be a more sensitive serum marker than CRP [12,14].

Unlike PCT, DNI, and CRP, the implications of SAA are under investigation. Moreover, its usefulness in patients with hematologic diseases, including hematologic malignancies, remains unclear. Therefore, this study aimed to evaluate and compare the diagnostic and prognostic performance of PCT, DNI, CRP, and SAA in patients with hematologic diseases.

## 2. Materials and Methods

### 2.1. Laboratory Assays

An automated chemistry analyzer was used to measure PCT, CRP, and SAA levels. The Atellica BRAHMS procalcitonin assay, a chemiluminescent immunoassay (CLIA)-based assay, was performed using the Atellica IM automated analyzer (Siemens Healthcare Diagnostics, Tarrytown, NY, USA) for PCT; the Roche C-Reactive Protein Gen.4, a particle-enhanced immunoturbidimetric assay, was performed using the Cobas c702 automated chemistry analyzer (Roche Diagnostics, Basel, Switzerland). The serum amyloid A kit, a latex-enhanced immunoturbidimetric assay (MedicalSystem Biotechnology Co. Ltd., Ningbo, China), was performed using the Cobas c702 analyzer (Roche Diagnostics). DNI was measured using an ADVIA 2120i automated hematology analyzer (Siemens Healthcare Diagnostics).

### 2.2. Study Design

From October 2021 to February 2022, 127 specimens were collected from patients who visited Wonju Severance Christian Hospital, a tertiary university-affiliated hospital located in Wonju, South Korea, for bone marrow examination and were diagnosed with hematologic diseases. In previous studies, the positive blood culture rate in patients with hematological diseases has been reported to be about 17–23% [16,17]. The minimal required sample size was 26 for the infected group and 104 for the uninfected (control) group, and it was calculated using G*Power software (latest ver. 3.1.9.7; Heinrich-Heine-Universität Düsseldorf, Düsseldorf, Germany) with effect size = 0.8, α = 0.05, β = 0.95, and an allocation ratio of 4.

Serum specimens submitted to our clinical laboratory for routine chemistry tests were collected after test completion. The remaining samples were aliquoted into three Axygen 1.7 mL MaxyClear Snaplock microcentrifuge tubes (Axygen Scientific Inc., Union City, CA, USA). Serum (0.5 mL) was collected from each microtube; samples were stored at −70 °C until the target number of samples was collected. On the same day, all specimens were thawed at room temperature, and each aliquot was analyzed using different assays (PCT, CRP, and SAA). DNI measurements were obtained within 24 h of blood sampling and performed using ethylenediaminetetraacetic acid-treated tubes.

Demographic and clinical data, including age, sex, body mass index, bone marrow aspiration result, comorbid diseases, vital signs, hospital visit (in/outpatient), treatment, and date of admission and death, were collected retrospectively by reviewing patients’ electronic medical records (EMRs). The body surface area was calculated using the Dubois formula. Laboratory data collected were as follows: white blood cell (WBC) count, hemoglobin, platelet count, absolute neutrophil count (ANC) for complete blood cell count (CBC), and erythrocyte sedimentation rate (ESR). Routine chemistry tests included the determination of albumin, aspartate transaminase, alanine aminotransferase, alkaline phosphatase, total bilirubin, blood urea nitrogen, creatinine, and estimated glomerular filtration rate (eGFR), along with microbiological data. The eGFR was calculated using the Modification of Diet in Renal Disease 4-variable (isotope dilution mass spectrometry traceable) formula. CBC results were determined using the ADVIA 2120i automated hematology analyzer (Siemens Healthcare Diagnostics), ESR was established using the TEST-1 analyzer (SIRE Analytical Systems, Udine, Italy), and routine chemistry results were determined using a Cobas c702 automated chemistry analyzer (Roche Diagnostics).

Definitions used in the present study were as follows: infection was defined as the presence of positive result from either blood or urine culture; positive culture result was defined when an inoculated plate produced >1000 CFU/mL (blood culture) or 10,000 CFU/mL (urine culture) of 1 or 2 organisms; only patients who had a positive result in either blood or urine culture were classified into the infection group. Cutaneous skin infection, an infection from a cause other than bacteria (e.g., candida), suspected contamination, etc., were excluded from the diagnosis of infection; neutropenia was defined as an ANC < 0.5 × 10^9^/L; fever was defined as an axillary temperature ≥ 37.5 °C [3,4].

### 2.3. Statistical Analysis

The patients were divided into the infected and uninfected groups. Data distributions are presented as frequencies and percentages for categorical data and compared using the chi-squared test. For numerical data, the Kolmogorov–Smirnov test was used to confirm normality, and a *p*-value > 0.05 indicated a normal data distribution (parametric data). For parametric data, results are presented as mean ± standard deviation (SD), with the Student’s *t*-test used for comparison. For non-parametric data, results are presented as medians and interquartile ranges, and the Mann–Whitney U test was used for comparison.

Each pair of assays was correlated using Spearman’s rank correlation, given that the results of all three assays were non-parametric, and the correlation coefficients were compared. Receiver operating curve (ROC) analysis was conducted to evaluate the clinical performance for diagnosing infection, and the value of the area under the curve (AUC) was compared. Cutoff values that showed the best sensitivity and specificity were calculated. The sensitivity, specificity, positive predictive value (PPV), negative predictive value (NPV), positive likelihood ratio (LR+), and negative likelihood ratio (LR−) were calculated according to cutoff values and some medical decision levels; the 95% confidence intervals of each item were also calculated. The performance of each marker, alone or in conjunction with other markers, was also compared in all patients and patients with neutropenia. The 90-day survival analysis was performed using Kaplan–Meier estimation, and statistical significance was confirmed using the log-rank test. Statistical analyses were performed using SPSS (version 25.0; IBM Corp., Armonk, NY, USA) and Microsoft Excel 2019 (Microsoft Corp., Redmond, WA, USA) with Analyse-it version 5.92 (Analyse-it Software, Ltd., Leeds, UK). Statistical significance was set at *p* < 0.05.

## 3. Results

### 3.1. Baseline Characteristics

Table 1 presents the baseline characteristics of the 127 included patients. The infected and uninfected groups comprised 27 (21.3%) and 100 patients, respectively. Of these patients, 51 (40.2%) were male, and 76 (59.8%) were female. The mean age of the patients was 65.5 ± 14.1 years. The most commonly diagnosed hematologic diseases were acute myeloid leukemia (*n* = 24, 18.9%) and lymphoma (*n* = 24, 18.9%), as determined by bone marrow assessment. Among underlying diseases, cardiovascular diseases were more frequent in the uninfected group (*p* = 0.0399), whereas renal diseases were more frequent in the infected group (*p* = 0.0260). The mean body temperatures in the infected and uninfected groups were 38.2 ± 0.2 and 37.6 ± 0.3 °C (*p* = 0.0010), respectively. Other vital signs were not significantly different between the two groups. Regarding laboratory findings, compared to the uninfected group, ESR and all four markers were markedly elevated in the infected group, whereas WBC and platelet counts and ANC were substantially reduced (*p* < 0.05). Neutropenia was present in 14 (51.9%) and 17 (17.0%) patients (*p* = 0.0002), and fever was noted in 27 (100%) and 72 (56.7%) patients in the infected and uninfected groups, respectively (*p* = 0.0014).

### 3.2. Comparison of PCT, DNI, CRP, and SAA

On segregating patients based on the presence or absence of neutropenia, PCT, DNI, CRP, and SAA levels in the infected group were significantly elevated in both patients with neutropenia (*p* = 0.0004, 0.0017, and *p* < 0.0001, respectively) and those without neutropenia (*p* = 0.0061, *p* < 0.0001, and *p* < 0.0001, respectively) (Figure 1). Spearman’s correlation coefficients are shown in Figure 2. Combining CRP and SAA afforded the best correlation (ρ = 0.727) among the examined markers.

### 3.3. Diagnostic Performance

The clinical performance for infection diagnosis was compared using the obtained AUC values, as shown in Figure 3. For all patients, SAA (*p* = 0.024) and CRP (*p* = 0.0290) showed considerably higher diagnostic performance than PCT. In patients with neutropenia, the diagnostic performance of the four markers did not differ significantly. However, in patients without neutropenia, SAA exhibited considerably higher diagnostic performance than PCT (*p* = 0.0018), CRP (*p* = 0.0451), and DNI (*p* = 0.0008). Table 2 shows the performance of PCT, DNI, CRP, and SAA for diagnosing infection at the cutoff values and medical decision levels. Cutoff values exhibiting both the best sensitivity and specificity for PCT, DNI, CRP, and SAA were 0.12, 4.5, 18.5, and 39.5 mg/L, respectively. Diagnostic performances at medical decision levels for PCT, DNI, CRP, and SAA were 0.20 and 0.50 ng/mL [18], 2.7 and 6.5% [8,9], 5.0 and 10.0 mg/L [19], and 10.0 mg/L [2], respectively. For SAA, performance was analyzed at 70% specificity. Table 3 presents the performance of each marker, alone or in conjunction with other markers, for diagnosing infection in all patients and those with neutropenia. The performance at the cutoff levels for each marker alone is described in Table 2. For all patients, SAA alone presented the highest sensitivity (82.1%) at the cutoff level. Compared with each marker alone, specificity was substantially increased with all combinations. In terms of sensitivity, the combination of CRP and SAA exhibited the highest sensitivity (75.0%). In patients with neutropenia, DNI showed the highest sensitivity (85.7%), whereas PCT presented high specificity (94.1%). Moreover, specificity increased by up to 100% for certain combinations.

### 3.4. Survival Analysis

Figure 4 presents the Kaplan–Meier survival estimates based on whether marker levels were elevated or normal, determined according to the cutoff values presented in Table 2. For all patients, only increased PCT levels significantly correlated with 90-day mortality (*p* = 0.0250).

## 4. Discussion

SAA is an APR with approximately 12-kDa-molecular weight [20] known to be produced increasingly in response to inflammation. SAA is primarily synthesized by hepatocytes in the liver but can be produced in several other tissues, including macrophages, kidneys, lungs, adipocytes, and smooth muscle cells [11]. It was first identified in the 1970s in a study assessing antibodies that cross-react with protein components in acute-phase plasma, known as an inflammatory-associated amyloid substance of approximately 104 amino acids. [21] Four genotypes of SAA have been identified (SAA1, SAA2, SAA3, and SAA4), all located on the short arm of chromosome 11 [2,22]. Among them, SAA1 and SAA2 are considered highly homologous and constitute APR forms of SAA [1]. SAA3 is considered a pseudogene, whereas SAA4 is deemed a constituent form [22]. Approximately 95% of liver-derived SAA has been associated with high-density lipoprotein (HDL) [23], and SAA reportedly binds to fraction 3 of HDL [24]. SAA plays a major role as an inflammatory mediator by enhancing the affinity of HDL for macrophages and has been associated with the metabolism and transport of cholesterol [25], adhesion and migration of inflammatory cells, and activation of neutrophils [26]. Reportedly, SAA levels begin to increase during the early phase of inflammation and reach up to 1000-fold in the blood at 24–36 h after the onset of acute-phase inflammation, decrease after 4–5 days, and return to baseline after 10–14 days [27]. Several methods have been developed to detect SAA levels. Enzyme-linked immunosorbent assays (ELISAs) or radioimmunoassay (RIA)-based SAA assays are highly sensitive but time-consuming and not automated. Immunonephelometry- or immunoturbidimetry-based assays are less sensitive than ELISA- or RIA-based assays; however, these assays utilize automated analyzers [2]. Herein, we performed an immunoturbidimetry-based SAA assay.

Fever is a frequently encountered complication during the progression or treatment of hematologic diseases, including hematologic malignancies; however, the underlying cause of fever is often non-specific and cannot be identified during the early phase, warranting special caution [4]. Infection remains one of the major causes of mortality and morbidity. In clinical practice, it is crucial to inform clinicians regarding the initiation and termination of antibiotic treatment by judging the patients’ potential for occult infection in real-world clinical settings [28]. During chemotherapy, patients with neutropenia are vulnerable to infection, and if fever is present, patients could be experiencing disease progression, underlying infection, or other causes [29].

In the present study, we evaluated PCT, DNI, CRP, and SAA levels in serum samples from 127 patients with hematologic diseases. Among them, 93 (73.2%) patients were diagnosed with hematologic malignancy (e.g., leukemia, lymphoma, and multiple myeloma), 22 (17.3%) had MDS/myeloproliferative neoplasms, and 12 had other hematologic diseases. Furthermore, the levels of all four markers evaluated were significantly higher in the infected group than in the uninfected group. On further segregating patients into those with and without neutropenia, patients with infection exhibited substantially higher levels of examined markers than those without infection. Patients with neutropenia with infection exhibited higher SAA levels than patients without neutropenia with infection. In a previous study [4], the neutropenia group had higher PCT and CRP levels than the non-neutropenia group, considering patients with hematologic malignancies. However, we found that the PCT and CRP levels were similar in patients with and without neutropenia with infection. The dynamics of SAA levels in neutropenia and non-neutropenia remain undefined; however, SAA is expected to exhibit increased discriminative power in patients with neutropenia.

In addition, we evaluated the correlation between pairs of markers. The results were consistent with those of previous studies, in which the correlation coefficient between PCT and CRP was 0.460 in patients with hematologic malignancies [4], and between CRP and SAA, it was 0.682 in patients with septic shock [15].

Furthermore, we examined the clinical performance for diagnosing infection. In the ROC analysis, SAA showed higher diagnostic performance than PCT in all patients and those without neutropenia. In patients with neutropenia, the AUC value of the DNI was the highest among the examined markers but was non-significant. Our findings revealed higher AUC values for diagnosing infections than those of previous studies. Yang et al. [3] reported that the AUC values of PCT and CRP were 0.651 and 0.566 for all patients, 0.624 and 0.500 for those with neutropenia, and 0.757 and 0.763 for those without neutropenia, respectively. Ebihara et al. [4] showed that the AUC values of PCT and CRP were 0.753 and 0.454 for all patients, and 0.746 and 0.556 in those with neutropenia, respectively. Batirel et al. [30] reported the AUC values of CRP and SAA as 0.72 and 0.68, respectively, in patients with neutropenia. Park et al. [31] reported an AUC value of 0.804 for DNI, while Seok et al. [9] reported it as 0.88 for detecting infection. In previous studies, patients with fever were selected separately; however, we included a few patients without fever, given that we used the remaining samples of patients who visited our hospital for bone marrow examination during the study period. In previous reports, fever was defined as a single oral temperature of >38.3 °C or a temperature of >38.0 °C for 1 h [28,30,32]. In the present study, fever was defined as an axillary temperature of ≥37.5 °C [3,4]. Given the differences in the experimental design, the present study may have simplified the discrimination of infected and uninfected patients and between infection fever, tumor fever, and fever of other origins.

Diagnostic agreements, such as sensitivity, specificity, PPV, NPV, LR+, and LR−, were calculated at the cutoff values and certain medical decision levels. Herein, SAA alone showed the highest sensitivity among the examined markers in all patients and those with neutropenia. The combination of CRP and SAA levels afforded the highest specificity. In previous studies, the sensitivity of PCT ranged between 33.3 and 100% at 0.1 ng/mL and between 21.0 and 92.9% at 0.5 ng/mL; the specificity of PCT ranged between 22.9 and 77.0% at 0.1 ng/mL and between 45.5 and 86.5% at 0.5 ng/mL [3,28,33,34]. For DNI, previous studies have reported a sensitivity and specificity of 73.4 and 97.9%, respectively, at a level of 2.7% [9] and 81.3 and 91.0% at 6.5% [8]. For CRP, sensitivity and specificity values of 79.8 and 29.3%, respectively, have been reported at 25 mg/L [3] and 66.7 and 46.6% at 50 mg/L [28]. Data regarding the diagnostic agreement of SAA in patients with hematologic diseases are limited. Sensitivity and specificity values of 39 and 81% for SAA were noted at a cutoff of 282.0 mg/L in a study predicting pneumonia in older adults [35], and values of 100 and 72.2% were detected at a 6.7 mg/L cutoff in a study predicting necrotizing enterocolitis in premature infants [36]. However, directly comparing these results with our present findings is challenging, given the different assay reagents and study designs employed.

Based on the Kaplan–Meier survival analysis, only the PCT levels were significantly associated with 90-day mortality. Previously, a PCT level > 0.46 ng/mL was shown to be associated with the occurrence of septic shock and death in patients with febrile neutropenia [37]. For DNI, a previous study reported an elevated DNI level of >20% to be positively correlated with mortality [10]; however, we used a cutoff point of 4.5%. Data regarding the relationship between SAA levels and mortality remain limited. Reportedly, the SAA level can be associated with the severity of acute pancreatitis [38]. Conversely, SAA was found to be a poor predictor of mortality in patients with septic shock [15]. Herein, the cutoff SAA value was 39.5 mg/L, which was substantially lower than the cutoff values documented in previous reports [35,38]. Large-scale follow-up studies of SAA are needed to assess prognosis and mortality.

The present study has a few limitations. First, we collected samples from patients who visited our hospital for bone marrow examination. Patients with fever or those with hematologic malignancies were not separately screened; instead, we included samples collected during the study period. We excluded patients with no abnormal findings on bone marrow examination. Second, we focused on comparing the four markers. We collected only 127 samples, which met the minimum number for method comparison according to the Clinical Laboratory Standards Institute (CLSI) guideline EP09-A3 [39]. Accordingly, additional studies with larger sample sizes are required. Third, we relied solely on electronic medical records (EMRs), as the IRB waived the need for informed consent. As we could only collect the remaining samples submitted to our laboratory after completing routine tests, direct patient selection was not possible. Further clinical studies are required to evaluate and validate the clinical usefulness and efficacy of SAA in clinical practice.

## 5. Conclusions

This study evaluated and compared the clinical utility of PCT, DNI, CRP, and SAA levels in diagnosing infection in patients with hematologic diseases. SAA exhibited comparable performance to PCT, DNI, and CRP and can be deemed a good biomarker for diagnosing infection in all patients and those with neutropenia. The SAA level could help distinguish the origin of fever, such as infection, tumor fever, or other causes, but is limited in terms of predicting prognosis. SAA is a valuable marker, and improved clinical performance could be expected when combined with PCT, DNI, and CRP. In conclusion, the SAA assay shows beneficial performance in routine testing.

## Figures and Tables

**Figure 1 diagnostics-13-01213-f001:**
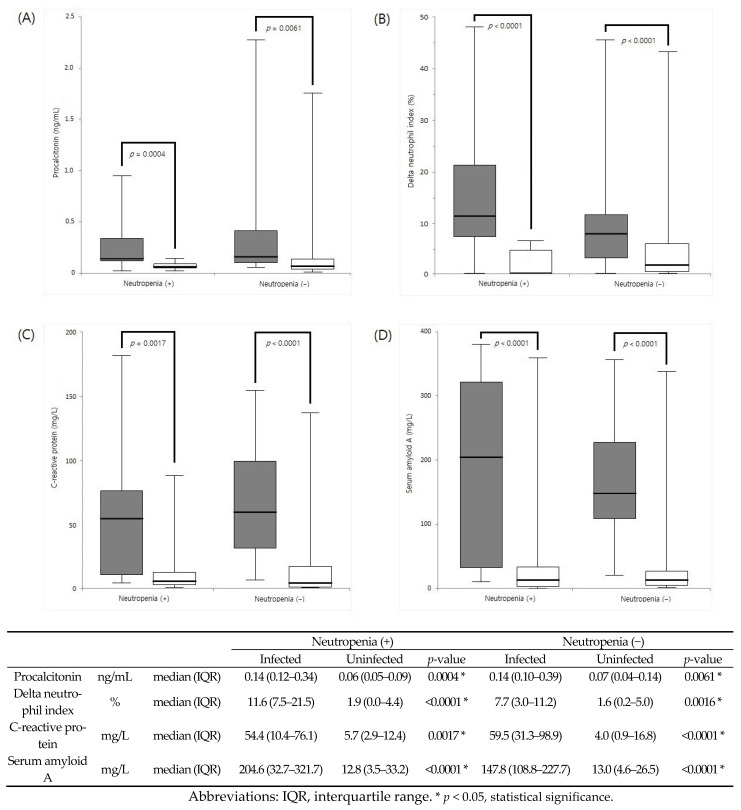
Comparison of (**A**) procalcitonin, (**B**) delta neutrophil index, (**C**) C-reactive protein, and (**D**) serum amyloid A between patients with and without neutropenia.

**Figure 2 diagnostics-13-01213-f002:**
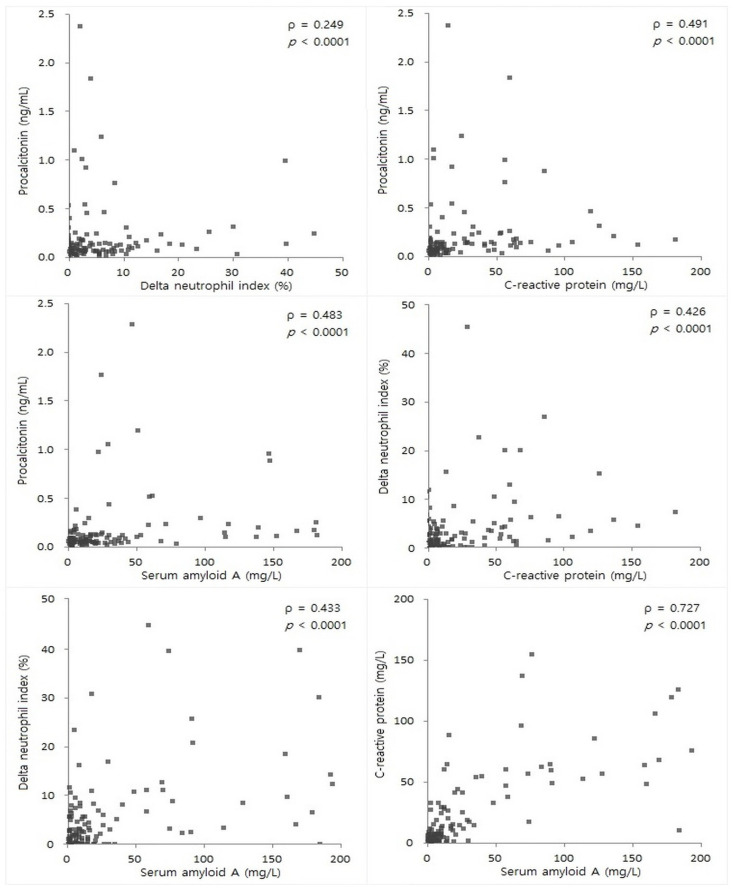
Correlations between the levels of each pair of examined markers among procalcitonin, delta neutrophil index, C-reactive protein, and serum amyloid A.

**Figure 3 diagnostics-13-01213-f003:**
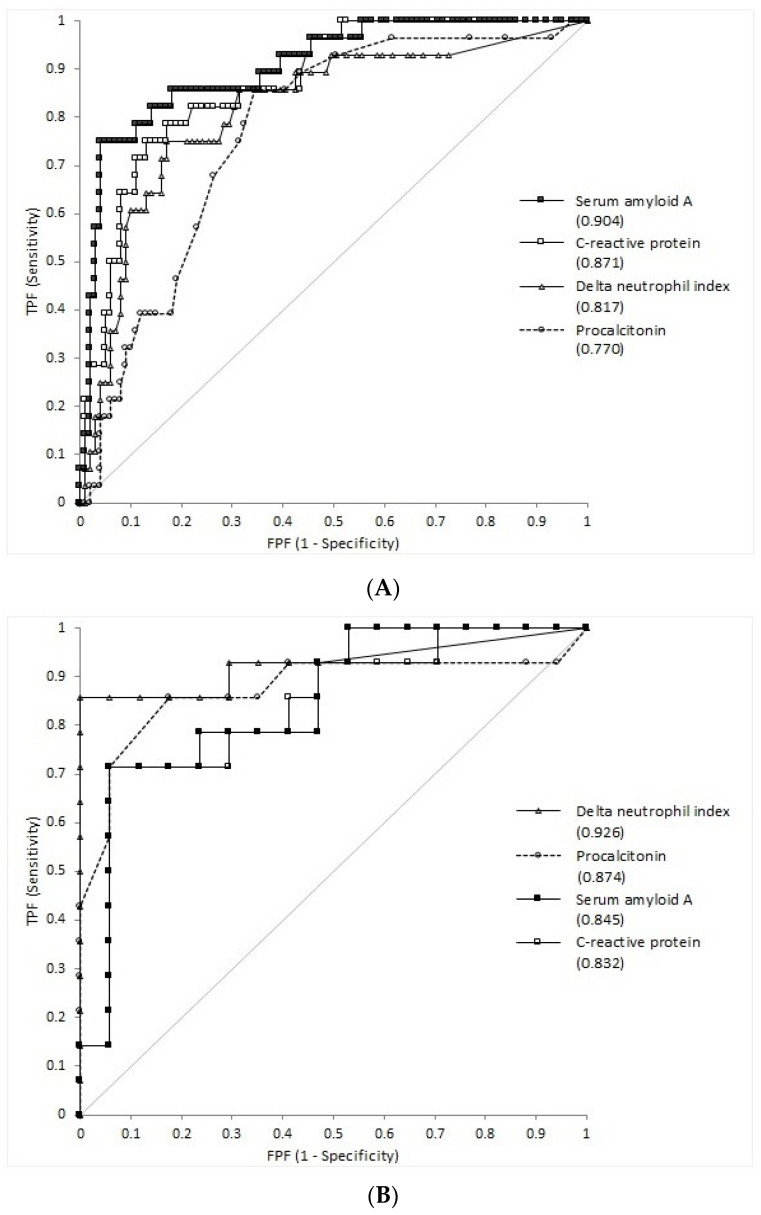
Receiver operating characteristic curves of procalcitonin, delta neutrophil index, C-reactive protein, and serum amyloid A for (**A**) all patients, (**B**) patients with neutropenia, and (**C**) patients without neutropenia.

**Figure 4 diagnostics-13-01213-f004:**
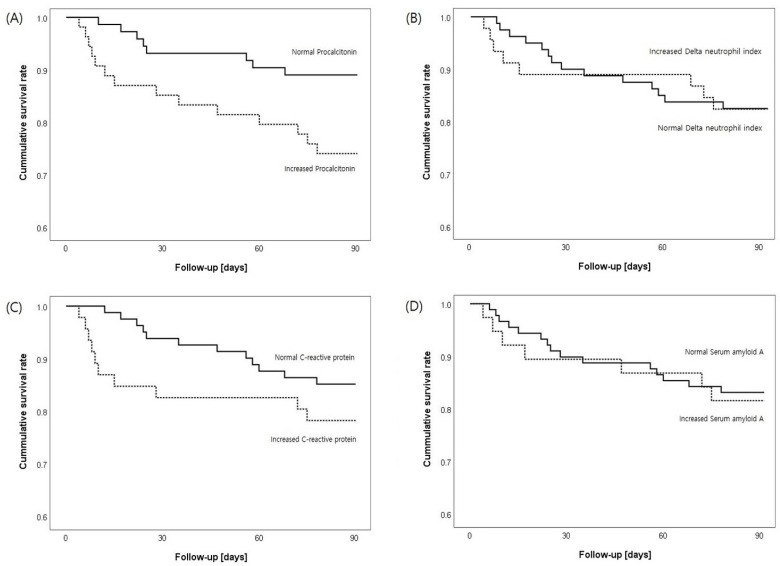
Kaplan–Meier survival estimates of patients by the levels of (**A**) procalcitonin, (**B**) delta neutrophil index, (**C**) C-reactive protein, and (**D**) serum amyloid A (mean and 95% confidence intervals).

**Table 1 diagnostics-13-01213-t001:** Baseline characteristics of patients (*n* = 127).

Characteristics		Units	Feature	Patients	Infected	Uninfected	*p*-Value
Baseline variables							
*n*			No. (%)	127 (100)	27 (21.3)	100 (78.7)	-
Age †		years	mean ± SD	65.5 ± 14.1	61.8 ± 15.9	66.6 ± 13.5	0.1231
Sex	Male		No. (%)	51 (40.2)	10 (37.0)	41 (41.0)	0.7093
	Female		No. (%)	76 (59.8)	17 (63.0)	59 (59.0)	
Body mass index †		kg/m^2^	mean ± SD	23.5 ± 3.7	23.9 ± 4.9	23.4 ± 3.4	0.5425
Hematologic disease							
Acute myeloid leukemia			No. (%)	43 (33.9)	11 (40.7)	32 (32.0)	0.3944
B/T lymphoblastic leukemia			No. (%)	5 (3.9)	1 (3.7)	4 (4.0)	0.9440
Chronic lymphoid leukemia			No. (%)	2 (1.6)	0 (0.0)	2 (2.0)	0.4589
Lymphoma			No. (%)	24 (18.9)	6 (22.2)	18 (18.0)	0.6190
MDS/MPN			No. (%)	22 (17.3)	3 (11.1)	19 (19.0)	0.3365
Multiple myeloma			No. (%)	19 (15.0)	3 (11.1)	16 (16.0)	0.5274
Aplastic anemia			No. (%)	5 (3.9)	2 (7.4)	3 (3.0)	0.2960
Hemophagocytic lymphohistiocytosis			No. (%)	3 (2.4)	1 (3.7)	2 (2.0)	0.6050
Other			No. (%)	4 (3.1)	0 (0.0)	4 (4.0)	0.2910
Underlying disease							
Hypertension			No. (%)	52 (40.9)	14 (51.9)	38 (38.0)	0.1940
Diabetes mellitus			No. (%)	23 (18.1)	4 (14.8)	19 (19.0)	0.6163
Cardiovascular disease			No. (%)	50 (39.4)	6 (22.2)	44 (44.0)	0.0399 *
Renal disease			No. (%)	31 (24.4)	11 (40.7)	20 (20.0)	0.0260 *
Hepatic disease			No. (%)	21 (16.5)	4 (14.8)	17 (17.0)	0.7862
Pulmonary disease			No. (%)	22 (17.3)	4 (14.8)	18 (18.0)	0.6980
Cerebrovascular accident			No. (%)	26 (20.5)	8 (29.6)	18 (18.0)	0.1839
Solid tumor			No. (%)	13 (10.2)	3 (11.1)	10 (10.0)	0.8658
Other			No. (%)	6 (4.7)	0 (0.0)	6 (6.0)	0.1922
Clinical findings †							
Systolic blood pressure		mmHg	mean ± SD	121.4 ± 15.2	117.4 ± 15.2	122.5 ± 15.1	0.1293
Diastolic blood pressure		mmHg	mean ± SD	72.8 ± 12.3	69.6 ± 13.7	73.7 ± 11.8	0.1179
Respiration rate		/min	mean ± SD	16.5 ± 2.8	17.1 ± 3.7	16.3 ± 2.5	0.1772
Pulse rate		/min	mean ± SD	83.7 ± 17.7	81.9 ± 16.8	84.2 ± 18.0	0.5548
Body temperature		°C	mean ± SD	37.7 ± 0.3	38.2 ± 0.2	37.6 ± 0.3	0.0010 *
Laboratory findings							
Erythrocyte sedimentation rate		mm/h	median (IQR)	12 (5–22)	21 (13–48)	9 (4–20)	<0.0001 *
White blood cell		×10^9^/L	median (IQR)	3.38 (1.33–8.05)	1.60 (0.41–5.93)	3.58 (1.80–8.37)	0.0080 *
Hemoglobin †		g/dL	mean ± SD	8.9 ± 1.8	8.4 ± 1.3	9.0 ± 1.9	0.1074
Platelet		×10^9^/L	median (IQR)	66 (34–165)	45 (33–83)	78 (36–172)	0.0169 *
Absolute neutrophil count		/μL	median (IQR)	2303 (576–5039)	431 (88–4002)	2347 (1116–5139)	0.0279 *
Albumin †		g/dL	mean ± SD	3.5 ± 0.5	3.4 ± 0.5	3.6 ± 0.6	0.1441
Aspartate transaminase		U/L	median (IQR)	18 (14.0–29.8)	20 (12.3–30.0)	18 (14.0–28.2)	0.9295
Alanine aminotransferase		U/L	median (IQR)	14 (11.0–25.8)	15 (11.0–20.8)	14 (11.4–26.0)	0.9741
Gamma-glutamyl transferase		U/L	median (IQR)	33 (20–51)	28 (17–39)	36 (22–58)	0.1462
Alkaline phosphatase		U/L	median (IQR)	85 (60–103)	85 (63–93)	85 (59–108)	0.6713
Total bilirubin		mg/dL	median (IQR)	0.55 (0.40–0.82)	0.66 (0.46–0.95)	0.51 (0.38–0.80)	0.1102
Urea nitrogen		mg/dL	median (IQR)	17.5 (11.2–24.1)	18.4 (13.5–25.4)	17.3 (11.1–23.6)	0.4297
Creatinine		mg/dL	median (IQR)	0.84 (0.61–1.04)	0.88 (0.56–1.57)	0.82 (0.61–1.00)	0.4541
Lactate dehydrogenase		U/L	median (IQR)	249 (175–296)	220 (165–263)	254 (188–311)	0.0587
Estimated glomerular filtration rate		mL/min/1.73 m^2^	median (IQR)	85.5 (56.7–117.3)	73.9 (33.4–116.7)	87.4 (60.4–117.9)	0.2214
Procalcitonin		ng/mL	median (IQR)	0.09 (0.05–0.16)	0.14 (0.11–0.28)	0.07 (0.05–0.13)	<0.0001 *
Delta neutrophil index		%	median (IQR)	2.5 (0.2–7.3)	9.5 (4.8–16.7)	1.5 (0.0–4.8)	<0.0001 *
C-reactive protein		mg/L	median (IQR)	6.9 (1.5–32.2)	56.4 (29.4–83.3)	4.9 (1.0–14.5)	<0.0001 *
Serum amyloid A		mg/L	median (IQR)	17.8 (5.5–52.9)	152.4 (99.7–307.3)	13.0 (4.1–28.6)	<0.0001 *
Others							
Neutropenia			No. (%)	31 (24.4)	14 (51.9)	17 (17.0)	0.0002 *
Fever			No. (%)	99 (78.0)	27 (100)	72 (56.7)	0.0014 *
Treatment	Chemotherapy		No. (%)	110 (86.7)	24 (88.9)	86 (86.0)	0.8899
	Stem cell transplantation		No. (%)	10 (7.9)	2 (7.4)	8 (8.0)	
Outcome	30-day mortality		No. (%)	13 (10.2)	5 (18.5)	8 (8.0)	0.1096
	90-day mortality		No. (%)	22 (17.3)	7 (25.9)	15 (15.0)	0.1831

Abbreviations: MPN, myeloproliferative neoplasm; IQR, interquartile range. * *p* < 0.05, statistical significance. † Some variables with a parametric distribution are presented as mean ± SD. Other variables with a non-parametric distribution are presented as median ± IQR.

**Table 2 diagnostics-13-01213-t002:** Performance of procalcitonin, delta neutrophil index, C-reactive protein, and serum amyloid A for the diagnosis of infection at various cutoff values (mean and 95% confidence intervals).

		Sensitivity	Specificity	PPV	NPV	LR+	LR−
Procalcitonin (ng/mL)							
>0.12		73.7 (64.3–81.4)	71.4 (52.9–84.7)	43.5 (33.9–53.6)	90.1 (83.4–94.3)	2.72 (1.78–4.05)	0.39 (0.21–0.65)
>0.20		39.3 (23.6–57.6)	86.9 (78.8–92.2)	45.8 (29.9–62.7)	83.5 (78.8–87.3)	2.99 (1.49–5.78)	0.70 (0.49–0.90)
>0.50		17.9 (7.9–35.6)	93.9 (87.4–97.2)	45.5 (21.5–71.7)	80.2 (77.2–82.9)	2.95 (1.00–8.36)	0.87 (0.68–1.00)
Delta neutrophil index (%)							
>2.7		85.7 (68.5–94.3)	63.6 (53.8–72.4)	40.0 (33.0–47.4)	94.0 (86.3–97.5)	2.36 (1.72–3.19)	0.22 (0.09–0.50)
>4.5		75.0 (56.6–87.3)	75.7 (65.4–82.3)	45.7 (36.0–55.6)	91.4 (84.6–95.3)	2.97 (1.96–4.42)	0.33 (0.17–0.59)
>6.5		67.9 (49.3–82.1)	83.8 (75.3–89.8)	54.3 (41.5–66.5)	90.2 (84.2–94.1)	4.20 (2.50–7.01)	0.38 (0.21–0.61)
C-reactive protein (mg/L)							
>5.0		96.4 (82.3–99.4)	50.5 (40.8–60.1)	35.5 (30.8–40.5)	98.0 (87.8–99.7)	1.95 (1.57–2.44)	0.07 (0.01–0.36)
>10.0		85.7 (68.5–94.3)	65.7 (55.9–74.3)	41.4 (34.1–49.1)	94.2 (86.6–97.6)	2.50 (1.80–3.42)	0.22 (0.09–0.49)
>18.5		78.6 (60.5–89.8)	80.8 (72.0–87.4)	53.7 (42.5–64.4)	93.0 (86.7–96.5)	4.09 (2.61–6.42)	0.27 (0.13–0.49)
Serum amyloid A (mg/L)							
>10.0		100 (87.9–100)	42.4 (33.2–52.3)	32.9 (29.3–36.8)	100 (100–100)	1.74 (1.50–2.10)	0.00 (0.00–0.29)
>22.4		85.7 (68.5–94.3)	70.7 (61.1–78.8)	45.3 (37.0–53.8)	94.6 (87.5–97.8)	2.93 (2.07–4.14)	0.20 (0.08–0.45)
>39.5		82.1 (64.4–92.1)	84.8 (76.5–90.6)	60.5 (48.3–71.6)	94.4 (88.3–97.4)	5.42 (3.33–8.95)	0.21 (0.09–0.42)

Abbreviations: PPV, positive predictive value; NPV, negative predictive value; LR+, positive likelihood ratio; LR−, negative likelihood ratio.

**Table 3 diagnostics-13-01213-t003:** Performance of each marker alone or in conjunction with other markers for the diagnosis of infection in all patients and patients with neutropenia (mean and 95% confidence intervals).

	Sensitivity	Specificity	PPV	NPV	LR+	LR−
All patients						
PCT	73.7 (64.3–81.4)	71.4 (52.9–84.7)	43.5 (33.9–53.6)	90.1 (83.4–94.3)	2.72 (1.78–4.05)	0.39 (0.21–0.65)
DNI	75.0 (56.6–87.3)	75.7 (65.4–82.3)	45.7 (36.0–55.6)	91.4 (84.6–95.3)	2.97 (1.96–4.42)	0.33 (0.17–0.59)
CRP	78.6 (60.5–89.8)	80.8 (72.0–87.4)	53.7 (42.5–64.4)	93.0 (86.7–96.5)	4.09 (2.61–6.42)	0.27 (0.13–0.49)
SAA	82.1 (64.4–92.1)	84.8 (76.5–90.6)	61.5 (49.5–72.3)	95.5 (89.4–98.1)	5.66 (3.51–9.28)	0.17 (0.07–0.37)
PCT + DNI	53.6 (35.8–70.5)	93.9 (87.4–97.2)	71.4 (51.7–85.4)	87.7 (82.7–91.4)	8.84 (3.88–20.26)	0.49 (0.31–0.69)
PCT + CRP	60.7 (42.4–76.4)	89.9 (82.4–94.4)	63.0 (46.8–76.7)	89.0 (83.6–92.8)	6.01 (3.13–11.52)	0.44 (0.26–0.65)
PCT + SAA	60.7 (42.4–76.4)	90.9 (83.6–95.1)	65.4 (48.6–79.0)	89.1 (83.7–92.9)	6.68 (3.39–13.20)	0.43 (0.26–0.64)
DNI + CRP	64.3 (45.8–79.3)	92.9 (86.1–96.5)	72.0 (54.5–84.7)	90.2 (84.8–93.8)	9.09 (4.33–19.34)	0.38 (0.22–0.59)
DNI + SAA	67.9 (49.3–82.1)	93.9 (87.4–97.2)	76.0 (58.3–87.8)	91.2 (85.8–94.7)	11.20 (5.10–24.99)	0.34 (0.19–0.54)
CRP + SAA	75.0 (56.6–87.3)	91.9 (84.9–95.8)	72.4 (56.6–84.1)	92.9 (87.2–96.1)	9.28 (4.73–18.56)	0.27 (0.14–0.47)
Patients with neutropenia						
PCT	78.6 (52.4–92.4)	94.1 (73.0–99.0)	91.7 (61.7–98.7)	84.2 (66.0–93.6)	13.36 (2.79–69.58)	0.23 (0.08–0.52)
DNI	85.7 (60.1–96.0)	82.4 (59.0–93.8)	80.0 (58.4–91.9)	87.5 (65.6–96.3)	4.86 (1.98–14.06)	0.17 (0.05–0.51)
CRP	71.4 (45.4–88.3)	88.2 (65.7–96.7)	83.3 (56.6–95.0)	78.9 (61.7–89.7)	6.07 (1.91–22.37)	0.32 (0.13–0.65)
SAA	78.6 (52.4–92.4)	82.4 (59.0–93.8)	78.6 (55.9–91.4)	82.4 (62.6–92.9)	4.45 (1.76–13.02)	0.26 (0.09–0.61)
PCT + DNI	71.4 (45.4–88.3)	100 (81.6–100)	100 (100–100)	0.81 (0.65–0.91)	-	0.29 (0.12–0.55)
PCT + CRP	64.3 (38.8–83.7)	100 (81.6–100)	100 (100–100)	77.3 (62.7–87.3)	-	0.36 (0.16–0.61)
PCT + SAA	64.3 (38.8–83.7)	94.1 (73.0–99.0)	90.0 (56.4–98.4)	76.2 (61.1–86.7)	10.93 (2.20–63.08)	0.38 (0.17–0.67)
DNI + CRP	64.3 (38.8–83.7)	94.1 (73.0–99.0)	90.0 (56.4–98.4)	76.2 (61.1–86.7)	10.93 (2.20–63.08)	0.38 (0.17–0.67)
DNI + SAA	71.4 (45.4–88.3)	100 (81.6–100)	100 (100–100)	0.81 (0.65–0.91)	-	0.29 (0.12–0.55)
CRP + SAA	71.4 (45.4–88.3)	100 (81.6–100)	100 (100–100)	0.81 (0.65–0.91)	-	0.29 (0.12–0.55)

Abbreviations: PPV, positive predictive value; NPV, negative predictive value; LR+, positive likelihood ratio; LR−, negative likelihood ratio; PCT, procalcitonin; DNI, delta neutrophil index; CRP, C-reactive protein; SAA, serum amyloid A.

## Data Availability

Data are available on request from the first author.

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
