# Peer review of "Performance Comparison of Procalcitonin, Delta Neutrophil Index, C-Reactive Protein, and Serum Amyloid A Levels in Patients with Hematologic Diseases"

_diagnostics, 2023, doi:10.3390/diagnostics13071213_

Round 1

Reviewer 1 Report (Previous Reviewer 3)

I continue to believe that there is no new information in the paper and thus I cannot support the publication of the paper.

Reviewer 2 Report (Previous Reviewer 2)

N/A

Reviewer 3 Report (Previous Reviewer 1)

After reviewing the corrected article, I still believe it may be of interest to readers.

This manuscript is a resubmission of an earlier submission. The following is a list of the peer review reports and author responses from that submission.

Round 1

Reviewer 1 Report

The manuscript may be a useful tool in the treatment of patients with acute myeloid leukemia, lymphomas, chronic lymphocytic leukemia, MDS/MPN, myeloma, and aplastic anemia. In all these groups, inflammatory conditions of varying severity occur during the course of treatment. The authors compare the known and used PCT and CRP markers with those less commonly used in the diagnosis of inflammation SAA and DNI. The obtained results are promising.

Reviewer 2 Report

The authors compared the diagnostic and prognostic performance of 
serum amyloid A (SAA), procalcitonin (PCT), delta neutrophil index 
(DNI), and C-reactive protein (CRP) in patients with hematologic 
diseases. They conclude that SAA exhibited comparable performance to 
PCT, DNI, and CRP and can be deemed a good biomarker for diagnosing 
infection in all patients and those with neutropenia.

The manuscript is interesting and well written. It could be considered for publication after some major revisions.

1. The infected group comprised only 27 samples in respect to the control group (100). Did the authors perform a sample size/power calculation? Please add it in the manuscript.

2. How the authors define the group of infected patients? How they exclude the presence of a possible infection in the uninfected group?

3. What are the infection diagnoses? Authors should briefly describe or possibly list the different types of infections.

Minor revision:

1. In the introduction (line 34-35: Inflammation has been associated....) is not present one or more references about the statement.

2. Tables and figures are clear and of good quality, but the text, abscissa and ordinate scale, should be slightly bigger.

Reviewer 3 Report

The study is of low importance as the described findings are very well-known, while the number of patients studied is very low and not uniform; different hematology disorders may influence the markers studied.